Integrating the RFID identification system for Charolaise breeding bulls with 3D imaging for virtual archive creation

Cappai Maria Grazia mgcappai@uniss.it 1
Gambella Filippo 2
Piccirilli Davide 2
Rubiu Nicola Graziano 3
Dimauro Corrado 4
Pazzona Antonio Luigi 2
Pinna Walter 1
1 Research Unit for Animal Nutrition, Department of Veterinary Medicine, University of Sassari , Sassari , Italy
2 Research Unit for Agriculture Engineering of the Department of Agriculture, University of Sassari , Sassari , Italy
3 NurEID Foundation , Nuragus , Italy
4 Research Unit for Animal Breeding Sciences of the Department of Agriculture, University of Sassari , Sassari , Italy
Bauzer Medeiros Claudia
Electronic publication date: 2019 Mar 4
Publication date: 2019
Volume: 5
Electronic Location ID: e179
Received 2018 Oct 4; Accepted 2019 Feb 8
Copyright: ©2019 Cappai et al.
Copyright year: 2019
Copyright holder: Cappai et al.
License: This is an open access article distributed under the terms of the Creative Commons Attribution License, which permits unrestricted use, distribution, reproduction and adaptation in any medium and for any purpose provided that it is properly attributed. For attribution, the original author(s), title, publication source (PeerJ Computer Science) and either DOI or URL of the article must be cited.
License URL: https://creativecommons.org/licenses/by/4.0/

Keywords: Electronic identification, Digital Image, Bull morphology, Data sharing, Stakeholder, 3D Digital Image

Funding: The authors received no funding for this work.

==============================
The individual electronic identification (EID) of cattle based on RFID technology (134.2 kHz ISO standard 11784) will definitely enter into force in European countries as an official means of animal identification from July 2019. Integrating EID with 3D digital images of the animal would lead to the creation of a virtual archive of breeding animals for the evaluation and promotion of morphology associated with economic traits, strategic in beef cattle production. The genetically-encoded morphology of bulls and cows together with the expression in the phenotype were the main drivers of omic technologies of beef cattle production. The evaluation of bulls raised for reproduction is mainly based on the conformation and heritability of traits, which culminates in muscle mass and optimized carcass traits in the offspring destined to be slaughtered. A bottom-up approach by way of SWOT analysis of the current morphological and functional evaluation process for bulls revealed a technological gap. The innovation of the process through the use of smart technologies was tested in the field. The conventional 2D scoring system based on visual inspection by breed experts was carried out on a 3D model of the live animal, which was found to be a faithful reproduction of live animal morphology, thanks to the non significant variance (p > 0.05) of means of the somatic measures determined on the virtual 3D model and on the real bull. The four main groups composing the scoring system of bull morphology can easily be carried out on the 3D model. These are as follows: (1) Muscular condition; (2) Skeletal development; (3) Functional traits; (4) Breed traits. The 3D-Bull model derived from the Structure from Motion (SfM) algorithm displays a high tech profile for the evaluation of animal morphology in an upgraded system.

Introduction

The current identification system for cattle in European Union countries will be modified starting from 18th July 2019, the date on which EU Reg 911/2004 will be implemented. The individual electronic identification (EID) of cattle based on radio frequency technology (RFID, 134.2 kHz) will become an official means of cattle identification, in addition to the double plastic ear tag identification system (EU Reg 1760/2000). The automatic RFID-based identification of cattle will be part of the innovation process of animal recording, encompassing a digitized and real time automatically generated database, within the so-called Precision Livestock Farming (PLF). In view of the potential offered by RFID technology, several perspectives may lead to advanced production systems, of particular importance both for regulation compliance and the profitable management of herds. In this scenario, production goals differ depending on whether dairy or beef cattle are considered. Modern beef cattle production relies on decades of animal selection, oriented to the improvement of breeding system efficiency, basically to reduce costs and increase profits. Bovine breeds display morphological differences relating to production goals, if dairy or beef cattle are considered. In particular, beef cattle breeds are selected to improve carcass worth and the evaluation of body morphology of the live animal encompasses the interplay between genetic selection, breeding practices and expression of desirable traits. In 2007, Nkrumah and coworkers correlated genetic and phenotypic behaviour based on economically relevant traits (ERT) in Angus and Charolaise breeds, including feed intake, feed conversion ratio, fertility and temperament as those with a direct impact on the management and sustainability of the beef cattle farming system. The genetic value of animals is meant to spread to the whole herd since the productivity and sustainability of the farm does not rely on the single individual. In view of this aspect, the very accurate selection of breeding bulls and cows by the farmer is carried out to optimise production performance, which represents the main economic driver of beef cattle herds. In light of recent research conducted worldwide and of the ever more sophisticated genomic techniques, the latest achievements by genetic investigations on beef cattle (Burrow & Dillon, 1997; Voisinet et al., 1997; Gibb et al., 1998; Sowell et al., 1998; Burrow & Corbet, 2000; Schwartzkopf-Genswein, Atwood & McAllister, 2003; Robinson & Oddy, 2004; Nkrumah et al., 2007; Moore, Mrode & Coffey, 2017 Su et al., 2017; Leal et al., 2018; Fonseca et al., 2018; Soulat et al., 2018) outline the need to obtain very specialized morphological lines of animals, with typical body measurements and breed conformation.

Several breeds of genetically selected beef cattle are acknowledged worldwide to be of elevated performance, and the Charolaise breed is one of these. This cattle breed originated in Bourgogne (France) and is characterized by the high quality of meat and acclimation characteristics (Briggs & Briggs, 1980). Nowadays, Charolaise beef cattle can be considered cosmopolite farm animals for meat production, along with some other breeds acknowledged worldwide to be selected for highly specialized traits. In Italy, the association of Charolaise and Limousine farmers (A.N.A.C.L.I.) was founded in 1987 with the creation of breed Registry Records at national level. The presence of the Charolaise breed in Italy is considerable and the Sardinia region contributes considerably to Italian beef meat production in terms of the number of heads raised (14,463 Charolaise heads; source A.N.A.C.L.I.: http://www.anacli.it/WEBSITE/index.php?pagid=2499sessione=).

Breeding bulls are selected by the farmer according to the type of herd and management. The most common type belongs to the “souche bouchér” line for the production of offspring with high live weight at birth and rapid skeletal and muscular development. Heifers or steers are normally slaughtered at an age of 16–18 months and weight to slaughter of about 650 kg.

The economic value of the breeding bull is strongly influenced by the morphological evaluation carried out in regional and national fairs for livestock, in which animals are scored by breed experts. As recently evaluated in a bio-economic model by Leal et al. (2018), the dressing percentage, associated with carcass weight at slaughter and relative yield, in Angus cattle is that with the largest economic value within herd productions. This concept may be applied to all beef cattle breeds.

It is important to underline that while advanced omic technologies have been used for decades in livestock sciences with particular regard to cattle, the evaluation of morphology through the scoring system of animals still conserves old-fashioned methods. The score attributed to the morphology of the breeding bull has an impact on the economic value of the animal and its progeny and increases the visibility of the farmer, who selects and raises animals for competitions. At present, different morphological sections are scored through an evaluation grid displaying a 2D schematic illustration of the generic bovine (Fig. 1). The score sheet reported in Fig. 1 displays the official scoring system adopted by the A.N.A.C.L.I. (2009). Breed experts judge animals in annual competitions on the basis of a semi-quantitative evaluation, based on the harmonic proportions of animal development, the morphology of anatomical districts and functional traits with direct impact on carcass composition. To a great extent, the overall evaluation is based on the experience of the breed evaluator and is carried out by classifying the animal according to: (1) Muscular condition; (2) Skeletal development; (3) Functional traits; (4) Breed specific traits. At present, this system appears to be technologically outdated if compared with the advanced technologies underlying the achievements of modern breed traits. In view of this gap, smart technologies largely adopted in the beef cattle sector (such as RFID, omic sciences, artificial insemination) have great potential for system improvement, as these may provide valid tools to implement the process for the evaluation of the breeding bull in a modern system. Against this backdrop, the goal of a competitive strategy relies on profitable and sustainable productions with regard to external competition (environmental or extrinsic competition) to strengthen the competitive internal advantages.

Figure 1 Actual 2D grid used for morphology scoring in beef cattle.

The scheme displays both somatometric measures and confomation scoring of beef cattle. Adapted from A.N.A.C.L.I. 2009 (http://www.anacli.it/WEBSITE/index.php?pagid=2437sessione=).

Precision livestock farming (PLF) has entered livestock farms relatively recently. The efficiency of management was to be improved by way of smart technologies in terms of several aspects of the everyday breeding practices of food producing animals (Banhazi et al., 2012; Berckmans, 2014; Fournel, Rousseau & Laberge, 2017), particularly of cattle. It was hypothesized that the contactless, real time and digitalized recording of animal data could be the starting point to both internal and external competitive evaluation of the Charolaise breeding bull. In particular, 3D imaging is a technology that is widely deployed in several sectors. In livestock, the deployment of the technology based on 3D digital imaging is in its infancy and the few papers on this topic testify to the cutting edge research exploring this new frontier of PLF (Wu et al., 2004; Kosgro, 2014; D’Eath et al., 2018). This paper describes an innovative and technological method that was explored to assess the potential of the integration of RFID for animal identification with 3D images of the real bull for the evaluation of the Charolaise breeding bull.

This research was undertaken with the aim of: (a) carrying out a SWOT analysis to identify the Strengths, Weaknesses, Opportunities and Threats of traditional vs. innovative processes of morphology evaluation; (b) testing the opportunity offered by the 3D digital model of the real animal in comparison to the 2D grid of the scoring system based on biometric measures in the perspective of a digital archive creation for e-commerce.

Material and Methods

Animal care and identification

The animals involved in this study were cared for according to European Union legislation on animal protection and welfare (EU Reg 2010/63/EU). The trial involved two Charolaise bulls from the same farm which were 3 and 4 years old, multi-champions of breed and category both at regional and national level. The two bulls enrolled in the trial possess high morphological and genetic value and serve as breeders for all the cows raised on the same farm (total heads raised excluding calves = 46). In addition, their semen is also sold for artificial insemination (AI) on other farms. Bulls on one farm can also serve several other farms, thus the impact on offspring is multiplied for the desirable traits that farmers intend to introduce to their own herd. Raising too many breeding bulls in the same herd would not be economically viable because AI can provide high profits with few, but highly selected animals. The choice to involve these two breeding bulls was therefore driven by conventional farm management practices on one hand and by their morphological value on the other.

Each bull was electronically identified with an endoruminal bolus (75 g. 70 × 21 mm RUMITAG bolus®) holding a passive HDX transponder (radio frequency 134.2 kHz, 32. 5 × 3.8 mm, ISO 11784-11785 Tiris 32 mm), on voluntary basis. In compliance with current European mandatory rules (EU Reg 1760/2000) for the individual identification of cattle, the bulls were also identified with double plastic eartags.

In vivo body measuring of the animals

Both bulls (body weight between 1.27 and 1.3 tons) were individually handled in two distinct moments and temporarily placed in a paddock with a flat concrete floor to allow the recording of body measurements. Wither’s height, trunk length and distance at thighs were taken with the Lydtin stick, on repeated measures until reaching the same value for three replicates. In this regard, it is necessary to highlight that, while stressful condition were kept to a minimum and the animals were familiar with the facilities and personnel, any sudden movement of the bull may require several repetitions to measure one single parameter until a repeatable and acceptable value could be safely achieved. All measurements were recorded and used to calculate sensitivity and accuracy of in vivo vs. 3D model measurements.

In field image capturing

The acquisition of 2D digital pictures of the animal was carried out in a geo-referenced system (Fig. 2), where the bull stood in the centre of a circle (r = 6 m) purposely drawn on the floor, in an area outside the barn, in natural daylight but in the shade of a shelter. The operator captured images with a digital camera integrated in a mobile phone (8 Mpx, focus length 4.15 mm, exposure 1/1721) and moved along the circumference where reference points of known dimensions were located. The choice to use an integrated camera in a mobile phone was made to test whether a device that is easily available could be suitable for this purpose.

Figure 2 Geo-referenced 360°—image capture system in farm.

The operator captured multiple consequent images with an overlap of at least 70%, moving around the bull that stood in the center of the circle.

The number of pictures taken from the different angles was determined on replicates until reaching the least frame capturing to obtain the best 3D textured model. The overlap extent of sequential images taken along the circumference was no less than 70%.

Post acquisition processing

The sets of images were elaborated with Agisoft Photoscan (Agisoft LLC ©, Russia) software, capable of performing photogrammetric processing of digital images and generating 3D spatial data, with the algorithm based on Structure from Motion (SFM). Through different consecutive steps starting from the chunk (the original digital image), the relative set of masks is created with the purpose of eliminating objects (masking the unnecessary objects of the picture) from the background (Fig. 3). The pictures are then aligned and a dense point cloud is generated (Fig. 4). Through the mesh of dense point clouds, the software finalizes the procedure through texturing and builds up the so called “Doll” model, from which the 3D model can be exported into different digital formats.

Figure 3 Series of masks generated from the chunks to eliminate background unnecessary objects.

Figure 4 Results from the dense point cloud of the Charolaise bull.

SWOT analysis of the process “as is” vs. “to be”

The bottom-up analysis of Strengths, Weaknesses, Opportunities and Threats was based on the potential of the introduction of the 3D image technology compared with the current scoring system for the beef bull.

• Definition of objectives

• Identification of users’ needs

• Strategies to enhance value

• Definition and improvement of services

SWOT analysis is based on intrinsic and extrinsic factors of the process that were analyzed as reported in Table 1.

Table 1 Scheme of the SWOT analysis carried out on the basis of intrinsic and extrinsic factors of the innovation of process for Charolaise bull morphology valorization.

SWOT analysis	Intrinsic factors	
		Strenghts	Weakness	
Extrinsic Factors	Opportunities	Development of strategies to increase opportunities	Minimization of threats to improve opportunities	
Threats	Exploit strengths to reduce threats	Planning of strategies of defense to reduce threats	

In this context, the opportunity offered by electronic identification was explored in the perspective of the creation of a virtual database where 3D images of bulls could be uploaded with the individual electronic identification code and the farm of origin. The implementation of the system with the introduction of such a new technological tool within the process would contribute to identifying the objectives to satisfy the concept expressed by Guatri (1991), according to whom the final goal of business relies on the capability of continuous auto-regeneration over time, with the sustainable creation of economic value.

In the light of the evaluation of the system “as is” (based on the current 2D evaluation score) and “to be” (after the development of the 3D model for digital archiving and digital data sharing) SWOT analysis was conducted to explore the introduction of 3D innovations to instrumental management for the evaluation of the breeding bull. This evaluation therefore focused on the specific extrinsic and intrinsic factors of the evaluation process to test whether the reduction of the technological gap on the farm (bottom) offers opportunities for and/or poses threats to the creation of economic value for farmers and, potentially, for stakeholders (scale-up).

Calculations, data analysis and statistics

A series of measurements for wither’s height, body length and rear trimness was carried out in vivo on each bull and on the respective 3D virtual model, until reaching the same value for three replicates.

The positive predictive value (PPV), sensitivity (or true positive rate, TPR), specificity (or true negative rate, TNR) and accuracy (ACC) of the biometric measurements determined on the 3D model in comparison with those retrieved from the real animal were calculated using the following formulas:

PPV=TPTP+FP×100;

TPR=TPFP+TN×100;

TNR=TNFP+TN×100;

Acc=TP+TNTP+TN+FP+FN×100,

Figure 5 Body measure taken on the 3D-bull model. Sequence of the topline (A), wither’s height (B) and rear trimness (C).

where TP is the number of virtual measurements matching with real measurements, FP is the number of different virtual measurements matching with real measurements, TN is the number of different virtual measurements differing from the real measurements, FN is the number of virtual measurements differing from real measurements.

The analysis of variance between means of the two series of measurements collected in vivo and on the 3D virtual model was carried out by ANOVA with SAS 9.2 (SAS Inst. Inc. Cary, NC). Results were considered statistically significant when p < 0.05.

Results

The least frame capturing required 81 images in this trial, with an at least 70% overlap between subsequent pictures on a 360° total capturing per bull. The operator moved around the animal in the geo-referenced system which allowed body proportions to be established. The process of image capturing took between 20′  and 22′  per bull. Somatic measurements collected in vivo and on the 3D-bull (Fig. 5) did not differ in a significant way. When comparing the two systems for taking somatic measurements the fact that no statistical significance between in vivo and virtual values was detected appeared highly encouraging for the adoption of the system in the field. Table 2 summarizes the results on 3D bull performance.

Table 2 Body measures of the live animal and respective 3D model.

In vivo and virtual values are reported as means and pooled-SD, express in meter. Statistic significance is set for p-value < 0.05.

Body measures (m)	in vivo	virtual	pooled-SD	p-value	
Topline	1.55	1.53	0.02	0.296	
Wither height	1.46	1.44	0.01	0.069	
Rear trimness	0.50	0.50	0.01	0.561	
Performance of the 3D bull					
TPR %		80			
TNR %		70			
PPV %		89			
Accuracy %		100			

In Table 3, results from the SWOT analysis are reported. On the basis of the series of measurements carried out both in vivo and on the 3D bull, the positive predictive value and the accuracy of the system turned out to range from 89% to 100% for the three sets of body measurements for wither’s height, body length and rear trimness.

Table 3 Scheme of the SWOT analysis emerged from the analysis of the process for Charolais bull morphology valorization.

SWOT analysis	Valorization of Charolais breeding bull morhology	
	Strengths	Weaknesses	
Intrinsic factors	• Competitiveness of bull genetic value
•  Experienced farmers
•  Worldwide market
•  AI diffused for semen trade
•  High dressing % of progeny
	•  Restrictions of animal movement
•  Costs of transport
•  Stressful conditions
•  Technology delay	
	Opportunities	Threats	
Extrinsic factors	•  Visibility of bull morphology
•  Generational change
•  Automation of operations
•  Limited investment to update the system
•  Virtual net of buyers/suppliers
•  E-commerce potentials	•  Aging population
•  Ethical trends on animal product consumption
•  Old fashioned scoring system
•  Climate change
•  Standardization of the system
•  Infectious disease	

Discussion

The opportunity offered by 3D digital imaging to carry out body measurements on a 3D-bull model improves the current system of morphological scoring considerably by objectifying evaluations. Indeed, the 2D grid with a schematic illustration of the generic bovine has to be filled in manually and does not allow any automation. On the other hand, the 3D bull is a faithful reproduction of the live animal, and is hence highly suitable for the morphological evaluation and appraisal of the genetic potential as a breeding bull through the phenotype.

Among the advantages offered by the 3D bull model, the collection of body measurements can be carried out in a safe way, much more comfortably for both the breed evaluator and the bull. In fact, fairs and the presence of other bulls during the scoring process may represent a stressful condition for the animal, which may react to the environmental stimuli in unpredictable ways. While expert personnel and animal friendly facilities are provided during fairs, safe conditions for operators and animal protection may be improved by the evaluation of the virtual model. Additionally, 3D digital images of bulls saved in purposely created virtual archives may ease access and the sharing of data among stakeholders. Such digital archives can greatly improve the visibility of the genetic (through the individual electronic code for genealogy) and phenotypic (morphological) value of the bull and potentially broaden the horizons of e-commerce if made available to a network of stakeholders. This opportunity opens up a series of considerations connected with different aspects of beef cattle management. When artificial insemination (AI) is used, the opportunity offered by the digital evaluation of bull morphology would allow the creation of a virtual archive for the e-commerce of semen from bulls with a high profile for functional traits.

In the SWOT analysis of the evaluation process of Charolaise bull morphology, we identified internal weaknesses in the costs of animal transport and animal welfare issues, both during transportation and exposition in dedicated fairs. In the “as is” process, the farmer must subscribe bulls to dedicated fairs both on regional and national levels, for which economic resources must be set aside to cover travel and subscription costs. In the “to be” process, the farmer will be able to subscribe animals to online virtual fairs, where animal morphology can be accessed by other remote subscribers and stakeholders thanks to the electronic code associated with the 3D model of the live bull. In this way, internal weaknesses can be minimized. The possibility of having a virtual archive where digital images of animals can be reached by stakeholders means that the animals do not have to be moved from the farm, hence offering substantial savings on travel costs. Reducing stress for the animals due to handling and their being loaded onto means of transport to reach unfamiliar settings is in agreement with other efficient solutions proposed by PLF (Banhazi et al., 2012; Berckmans, 2014) to optimise breeding and management practices on site. However, as the system is somewhat advanced, some stakeholders may be unready for the technology and this may lead to a delay in the standardization of the 3D-model based system.

Electronic identification through the RFID technology represents both a sound and reliable method for animal identification and a very promising system for the implementation of a virtual archive of recorded animals through the electronic digital number of the transponder, as observed in other filiéres (Cappai et al., 2014; Cappai et al., 2018; Cappai, Rubiu & Pinna, 2018). In the case of the electronically identified 3D bull, the record was implemented with 3D virtual images. The 3D-bull model for morphological evaluation through a scoring system has a series of indirect advantages, related to both animal health and welfare. The evaluation of animal morphology raised on farms can be promoted also when sanitary restrictions to the movement of animals are in force. As an external threat considered in the SWOT analysis, the presence of infectious diseases may impact negatively on animal movements outside the farm. For instance, in the case of Blue Tongue positivity in sheep of a given region, cattle movements are also restricted as bovines are a natural reservoir of the virus, despite not being clinically involved. Thus, participation in fairs, as well as movement for trading are forbidden, with consequent profit losses. The SWOT analysis pointed to different opportunities offered by the introduction of the 3D model for the evaluation of bull morphology oriented to an up-to-date competitive strategy for the sector. In fact, the analysis of the process “as is” and “to be” clearly highlights how the digital 3D model performance on the basis of the PPV can implement the evaluation system in a smart, contactless and automated way, which is easily shareable and accessible, whereas the 2D scoring system does not. The threat of external factors may be the most difficult aspect to deal with. So-called red meat consumption may decrease in the future, unless appropriate marketing strategies and adequate public information are prompted. This threat was considered in the SWOT analysis due to the increasingly aging population and their choice to prefer so-called “white meat” (that of poultry and rabbit). In addition, climate changes may pose the question of environmentally sustainable farming systems, with a potential contraction of meat consumption. Finally, ethical movements against animal product consumption may also influence market choices.

Conclusions

The results obtained from testing the feasibility of an innovative and technologically advanced methodological approach based on the implementation of an RIFD and 3D imaging system aimed to provide a general overview of valuable opportunities offered by the upgrade of the current system. The instrumental evaluation of bull morphology of the Charolaise breed in a completely digital 3D image successfully reflects that of the morphology of the live animal. The upgrade into a high-tech system for the evaluation of the breeding bull shows several potential applications as illustrated in the SWOT analysis leading to the innovation of the process. In perspective, the opportunities offered by such an innovative methodological approach may lead to the scale up of the integrated system based on individual RFID identification along with that of 3D digital image of the bull. In fact, the model has the potential for virtual archive creation and the experimental approach appears highly encouraging for further work to check scalability to a large number of animals.

Supplemental Information

Supplemental Information 1 Raw data expressed as mean of means from in field (in vivo from the live animal) and in lab (virtual measure on animal model) collection of somatic measures of Charolaise bulls

Click here for additional data file.

Supplemental Information 2 Bull video around the virtual space created in the farm

Vision of the 3D bull model on the basis of real body measures collected from the live animal in a virtual spatial motion.

Click here for additional data file.

Supplemental Information 3 Masks of the Bull

Different masks of the bull during model creation.

Click here for additional data file.

Supplemental Information 4 Body measures of 3D bull model

Backline measure of the 3D bull model. Virtual measures do not differ in a significant way from the real live animal.

Click here for additional data file.

The authors would like to thank Mr. Michele Filigheddu for his cooperation during fieldwork activities on his farm and for providing basic production data. The authors are also thankful to Dr. Santino Cherchi for his help during the study. The authors express their gratitude to A.N.A.C.L.I.

Additional Information and Declarations

Competing Interests

Author Contributions

Data Availability

Nicola Graziano Rubiu is the administrator and technical Director of NurEID. The authors declare there are no competing interests.

Maria Grazia Cappai conceived and designed the experiments, prepared figures and/or tables, authored or reviewed drafts of the paper, approved the final draft.

Filippo Gambella conceived and designed the experiments, contributed reagents/materials/analysis tools, performed the computation work, approved the final draft.

Davide Piccirilli performed the experiments, prepared figures and/or tables, performed the computation work, approved the final draft.

Nicola Graziano Rubiu analyzed the data, authored or reviewed drafts of the paper, approved the final draft, analysis of process and SWOT analysis.

Corrado Dimauro performed the experiments, analyzed the data, approved the final draft.

Antonio Luigi Pazzona contributed reagents/materials/analysis tools, performed the computation work, approved the final draft.

Walter Pinna contributed reagents/materials/analysis tools, authored or reviewed drafts of the paper, approved the final draft.

The following information was supplied regarding data availability:

Database of raw biometric measures from the live animal and the 3D model are available in the Supplemental Materials.

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
