# Peer review of "Integrating the RFID identification system for Charolaise breeding bulls with 3D imaging for virtual archive creation"

_PeerJ Computer Science, doi:10.7717/peerj-cs.179_

## Round 0.1 · original submission · Major Revisions

Though the overall approach is interesting, the paper lacks a description of the methodology used, and thus reproducibility of the results. The authors might consider concentrating into a few aspects rather than covering everything. As it is, no subject is treated with the depth required of a journal paper. In particular, authors should better describe the SWOT matrix, and show how the approach can scale up in practice to more animals. Authors should also describe how the proposal was validated in practice.

All reviewers comment on the fact that the sample is not big enough to validate the proposal. This is a crucial point, since otherwise the method cannot be automated, and thus not applicable in large scale.

Last but not least, related work needs to be updated with more recent references

·

Basic reporting

Paper presents a proposal of integration between electronic identification (EID) and 3D Digital Image of animal. Authors developed a SWOT analysis of actual bull morphological and functional evaluation (2D). Then they carried out a 3D model of the living animal, which presents to be adherent to live animial morphology.

Experimental design

The development of a SWOT analysis requires a deep understanding of the subject. Although authors presents a good literature review, the SWOT lacks of a methodological approach. It is important to describe how the proposed statements of the SWOT analysis were collected and validated.

Section 2.6 (calculation, data analysis and statistics) presents the performance of the proposed technique. Authors could describe more about how such metrics were calculated and what is a 'match' between virtual measures and real measures. It is also important to provide an analysis based on TPR and TNR. Is there any recall issues in the results?. Present a table summarizing the results can be a good way for clarification.

Validity of the findings

Results are presented at the paper, but authors could improve the discussion, presenting detailed statistic.

Conclusion are well stated and linked to original research question.

Additional comments

The paper is well written but you could improve the details of the applied method and presents a more descriptive result analysis. It is recommended to present the user's needs prior to the S.W.O.T analysis results. You could also describe (even with the help of a picture) how you conducted the series of measurements.

Reviewer 2 ·

Basic reporting

The article shows interesting technical to measure morphology in beef cattle.
Introduction and backgrounds are clear to show the context. Tables and figures are relevant and necessary to show the experiment and the results.
I think that material could be better explained and also, the analysis. The results could be better explored, including more tables with detailed calculations for PPV, TPR, TNP and acc.

Experimental design

The authors could explain better the SWOT matrix and why it was used in the context of the article, because it is not clear.

The authors should explain better the analysis of variance and in detail, what were the series of measurements. Did you use two bulls and how many measures by each one?

I consider that there was a small number of animals used in the experiment (only two bulls) a to test the 3-D against the standard method. Could you explain better this point of view?

Validity of the findings

The authors should explore better the results of data analysis, mainly the calculations of PPV, TPR, TNP and ACC, showing these results in a table.

I have doubts about the necessary number of animals to validate such methodology. Only two bulls are enough? P

The implementation of 3-D images has many advantages, but only in an automated system of evaluation. Mainly to the animal breeding application, that is necessary to measure in many animals.

Additional comments

The authors presented an interesting methodology to show alternative ways of measuring animals. 3-D images could be implemented in some official system to evaluate animals, as long as its implementation becomes easy, cheap and automated.

Reviewer 3 ·

Basic reporting

*** The literary references are not appropriate to the context. Lines 60-67 report "recent research efforts ongoing worldwide" on "sophisticated genomic techniques", but the most recent reference is from 2007.

*** The results should be better presented (section 3).

Experimental design

*** I am not sure if I could be replicate the experiment. Methods do not describe with sufficient detail and information to replicate. The methodology description (section 2) is short and show few details.

Validity of the findings

*** The discussions (section 4) need to be enriched, since they have little basis and some opinions derived from empirical observations. For this reason, the conclusion (section 5) is not appropriate because it does not provide solid elements of the discussion of the results.

Additional comments

1) The improvement obtained was not mentioned in the abstract. The abstract should make mention (at least) of the improvement obtained;
2) The authors used clear, technical and unambiguous writing;
3) The manuscript presents professional structure and auxiliary elements (figures, tables etc.).
4) The necessary definitions have been included and they are clear;
5) The research problem is aligned with the scope of the journal.
6) The research question is not "a question" because it is wide. The question represents a improvement in the solution of a problem, which is relevant and meaningful. The propositions introduce how the study contributes to filling the gaps of problem;
7) Data are controlled and the statistics used are clear and effective, though simple.

---

## Round 0.2 · Minor Revisions

Your remarks and the accompanying revised version show that you met the main problems raised by the reviewers. However, the paper still requires a few additional modifications to be published, namely

a) Your cover letter claims that your paper is not a complete experiment; rather, it presents a new methodology that may be scaled up under appropriate conditions. The revised version does not state this claim. The reader is still under the impression that you consider that an experiment carried out with 2 bulls is sufficient to validate your approach. Thus, please revise introduction and conclusions to point out that this is a description of a methodological approach, and that it was carried out with just two bulls to show its feasibility, but that further work is required for effective scaling up

b) There are many sentences that need revision, either because of grammar issues, or because there is something missing. This paper needs a thorough revision to correct grammar mistakes, but most importantly fix strange sentences. I strongly suggest you ask a native English speaker to help in this task. I point out three sets of lines as examples of text that is very hard to understand. Interestingly enough, most of the bigger problems occur in the new text introduced, so perhaps this was performed without as much thoroughness as when writing the first version.

lines 85-88 of the revised version "As recently evaluated in a bio-economic model by Leal and co-workers (2017), the dressing percentage, associated with carcass weight at slaughter and relative yield, in Angus cattle displayed to have the largest economic value within herd productions. Such concept may be applied to all beef cattle breeds. "

LInes 148-152: ". At this regard, it is necessary to highlight that, despite any stressful condition was minimized and animals were familiar with the facilities and personnel, any sudden movement of the bull may require several repetitions to measure one single parameter until a repeatable and acceptable value could be safely achieved. "

lines 241-242 and other subsequent lines "Among the advantages possessed by the 3D bull model, the collection of body measures can be carried out in a safe way, much comfortably for both the breed evaluator and the bull."

---

## Round 0.3 · Minor Revisions

You have met all the requirements. However, you still do not stress that this is an initial experiment and that it needs further studies and experimentation for actual scaling up and economic usefulness. Please add this clarification either at the introduction or as a final sentence to the conclusions.

Secondly, I strongly suggest you change the title. "e-commerce" only appears as an aside, and is not at all mentioned in the paper, except as motivation. Therefore, it should be eliminated from the title - for instance, a much better title could be "Integrating the RFID identification system for Charolaise breeding bulls with 3D imaging for virtual archive creation" - this is exactly what the paper does, and can be used in many other contexts beyond e-commerce (which limits the applicability of the research).

Thus, the paper can be accepted if

(a) One sentence is added to the end that this was conducted as an experiment, and that further work will be needed to check scalability to a large number of animals.
(b) "e-commerce" is eliminated from the title.

---

## Round 0.4 · accepted · Accept

Authors have complied with all requests, and the paper can be published as per the latest version (the new title, and conclusions indicate this needs further work for scalability)